# Standards Setting in Religious Education: Addressing the Quality of Teaching and Assessment Practices

Antonella Poncini [1,2]

1    Catholic Education Western Australia Limited, West Leederville, WA 6007, Australia;
     antonella.poncini@cewa.edu.au
2    School of Education, The University of Notre Dame Australia, Fremantle, WA 6160, Australia

**Abstract:** This paper describes a set of educational initiatives in Religious Education (RE). The paper addresses RE as an academic learning area and reserved for Kindergarten to Year 10 classrooms in Catholic schools. The initiatives described in the paper are evidence-based; they are guided by research and student achievement data addressing contemporary assessment reforms within the Australian educational context and considering the implications that the reforms have on the quality of student learning in RE. The reforms include quality assurance measures such as large-scale, standardised assessments (LSAs) and social moderation tools. The purpose of the measures is to establish standards that support the alignment of teaching and assessment practices. Such an alignment in the RE learning area has the potential to strengthen and improve student, as well as teacher, knowledge and understanding of the Catholic Faith Tradition. The paper draws from the Catholic RE school curriculum in Western Australia (WA), assessment principles outlined for schools in WA, an Australian Research Council Linkage project and local research about teacher perceptions of an LSA in RE. The paper supports local and national policy expectations recommending a review of the quality of RE as a learning area. Furthermore, given the prominence of the Catholic Church's mission of evangelisation and the nature and role of RE in Catholic schools as part of that mission, the paper endorses the Church's current stance on thoughtful dialogue and renewed efforts to bring a better understanding about Catholic culture and education to the world.

**Keywords:** religious education; curriculum pedagogy; assessment; measurement; quality assurance; moderation; judging standards; student achievement; reporting; whole-system approach; kindergarten to year 10

## 1. Introduction

Catholic schools are benefactors of student learning and growth. In Catholic schools, Religious Education (RE) is the priority learning area that should stand out (Congregation for Catholic Education (Congregation for Catholic Education (CCE) 1977, paras. 71, 72; Congregation for Catholic Education (CCE) 1988, paras. 66, 73), but always in "harmony with the formative aims" of the other learning areas (Pontifical Council for the Promotion of the New Evangelisation (Pontifical Council for the Promotion of the New Evangelisation 2020, para. 309). Therefore, at a time when quality assurance measures are being developed and implemented across core learning areas offered in Australian early years, primary and secondary schools, including Catholic schools (Department of Education and Training 2018; Masters 2018), no less should be expected of the RE learning area (Libreria Editrice Vaticana 1983, canons 801, 806).

The purpose of this paper is to describe educational initiatives specific to the RE learning area but not only unique to RE. The educational initiatives are designed to help set whole-system teaching, assessment, and reporting standards in RE. The aim of the initiatives is to benefit the quality of student learning in RE within and beyond the local context. The intended outcome of the initiatives endorses the educational pursuit for

excellence in RE and the Church's renewed attention to its missionary identity that calls for "greater capacity for sharing, communication and encounter, so as to journey together on the path of Christ and in docility to the Spirit" (Pontifical Council for the Promotion of the New Evangelisation 2020, para. 305).

The educational initiatives described in this paper propose a holistic and multidisciplinary approach to the implementation of the RE curriculum and engage in evidence-based teaching and assessment practices outside the scope of traditional and confessional RE classrooms. The initiatives are quality assurance measures that have been recently used to support the implementation of the RE learning area in Catholic schools in Western Australia (WA). These initiatives are consistent with measures used in other learning areas in WA schools. The first of the measures is a long-standing, large-scale assessment called *'The Bishops' Religious Literacy Assessment'* (BRLA), introduced by Catholic Education Western Australia (CEWA) in 2007. Other measures include judging standards in RE and associated exemplars of student achievement in RE; both are products of an Australian Research Council Linkage project LP 180100046 entitled, 'Improving teacher assessment capabilities using exemplars and online moderation' (Adie 2022). The research project focuses on using moderation opportunities based on scaled exemplars. The project, together with local research about teacher perceptions of the BRLA (Poncini 2018, 2021), elevates the value of the educational and religious awareness of student learning in RE. Furthermore, the initiatives presented in this paper build on scholarship focused on student learning and the progression of learning over time, by targeting the development of teacher content knowledge and teachers' capacity to design quality assessments (Andrich and Marais 2019; Gardner 2012).

CEWA acknowledges that RE is and should be a priority learning area (Catholic Education Commission of Western Australia (Catholic Education Commission of Western Australia (CECWA) 2009). RE has a significant role to play within the context of Catholic schools and within the broader context of the evangelising mission of the Catholic Church (Congregation for Catholic Education (CCE) 1988, 2014). However, for RE to fulfil its responsibility in the evangelising mission, religious educators need to be supported in terms of understanding the nature, role and prominence of the RE learning area (Congregation for Catholic Education (CCE) 2014, 2022; Vatican Council II 1965). Equally so, religious educators need to be directed to closely follow the teaching and assessment principles underpinning RE, and how these principles are aligned to other learning areas developed and implemented for schools in WA (School Curriculum and Standards Authority (SCSA) 2022a). At the local level, the educational initiatives described in this paper build on the whole CEWA priority that promotes quality Catholic education, and the professional development and formation of religious educators as part of that priority (Catholic Education Western Australia (CEWA) Limited 2022). Beyond the local level, the initiatives promote the ongoing dialogue and collaboration needed between classroom teachers and school and system leaders about teaching and assessment practices in RE that focus on the quality of student learning (National Catholic Education Commission (NCEC) 2018).

## 2. The Religious Education Curriculum for Catholic Schools in Western Australia

In Catholic schools in WA, RE is identified as the most important academic learning area reserved for the classroom (Catholic Education Commission of Western Australia (CECWA) 2009, para. 62). The RE curriculum comprises of content closely linked to the Catechism of the Catholic Church and supporting pedagogical and assessment resources. It is "both a learning area and an integral part of the educational philosophy and structure of the Catholic education system in Western Australia" (Hackett 2006, p. 36).

As a learning area, RE is implemented state-wide, across classrooms in Catholic primary, secondary and composite schools located in the Broome, Bunbury and Geraldton Dioceses and the Perth Archdiocese. The RE curriculum is developed by system-based educators working for CEWA on behalf of, and promulgated by, the local WA Conference of Catholic Bishops (Catholic Education Commission of Western Australia (CECWA) 2009,

para. 62). As such, the RE curriculum draws from the evangelising mission of the Catholic Church. This mission is about the transmission of the "Good News" or Gospel message of Jesus Christ (Francis I 2013; Paul VI 1975). Therefore, the Gospel message and the Christian response to that message is central to the development and implementation of the RE curriculum (Pontifical Council for the Promotion of the New Evangelisation 2020). A holistic approach to the transmission of the Gospel message is adopted. This approach is about nurturing faith through education (Montessori et al. 1965), and about the integration of that faith into the culture and life of every member participating in RE as a learning area (Congregation for Catholic Education (CCE) 1988, para. 66). Within this Catholic school perspective, the RE curriculum addresses the needs of students and provides students the opportunities to reflect on their personal experiences of a life in faith (Congregation for Catholic Education (CCE) 1988, para. 69) and to critically dialogue with the world around them in the same way as they potentially do with other acquired knowledge (Congregation for Catholic Education (CCE) 1988, paras. 58, 60, 62; Congregation for Catholic Education (CCE) 2014).

The nature and role of the RE curriculum for primary, secondary and composite schools within CEWA is aligned to the Catholic Church's policy expectations relevant to the treatment of RE as a learning area. Equally so, the RE curriculum for Catholic schools in WA is attributed to Bishop Gerard Holohan (Pendal 2008) and his interpretation of divine Revelation (Holohan 1999). The interpretation is not unique to the local context, and neither is the expectation that RE be treated as a priority, with the same systematic demands and rigour as other academic learning areas (Congregation for Catholic Education (CCE) 1977, paras. 71, 72; Congregation for Catholic Education (CCE) 1988, paras. 66, 73). For example, it is universally expected under the guidance of diocesan bishops that RE be promoted through "proper" educative processes (Libreria Editrice Vaticana 1983, canons 801, 806). Furthermore, the formal policy requirements issued by the Catholic Church demand RE to be delivered by "professional" Catholic schools and by a level of competency from teachers of RE who demonstrate "outstanding in correct doctrine, the witness of a Christian life, and teaching skill" (Libreria Editrice Vaticana 1983, canons 796–806). Therefore, the Catholic Church broadly and CEWA specifically recognise RE as more than an academic learning area and acknowledge that Catholic school leaders, together with classroom teachers of RE, have a significant role in effectively implementing this priority learning area within a holistic approach (Congregation for Catholic Education (CCE) 1988; Cullen 2019; Scott 2019).

The structure of the RE curriculum is consistent with the curricula of other Catholic education systems in Australia and modelled specifically on the Kindergarten to Year 10 curriculum that is legislated for implementation by government and non-government schools in WA (School Curriculum and Standards Authority (SCSA) 2022a). The students enrolled are aged approximately four to eighteen. The implementation of the RE curriculum in Kindergarten to Year 10 classrooms is specifically alongside (but not as an accessory) eight other important learning areas, and as such should be endorsed by school leaders and taught by classroom teachers with the same systematic demands and rigour afforded to the other learning areas (Catholic Education Commission of Western Australia (CECWA) 2009, para. 62). The learning areas other than RE that are mandated in schools in WA are: English, Mathematics, Science, Humanities and the Social Sciences, Health and Physical Education, The Arts, Technology and Languages (School Curriculum and Standards Authority (SCSA) 2022a).

It is, therefore, relevant, reasonable, and appropriate to consider and apply in RE the educational reforms promoted by government authorities and targeted for those other learning areas. In recent times, the educational reforms adopted by Australian educational jurisdictions, sectors and systems have been standards-based and assessment-focused (Masters 2018). The purpose of the reforms is to establish whole-system approaches to teaching and assessment practices that better support the quality of student learning and progress that learning (Department of Education and Training 2018).

To help support classroom teachers and school leaders of RE working in CEWA to better understand the nature, role and prominence of the RE learning area, the Religious

Education and Faith Formation (REFF) Team at the Catholic Education Office of Western Australia (CEOWA) has developed educational initiatives in the form of standardised teaching and assessment practices and resource materials. The process for development is aligned to the Strategic Directions for CEWA 2019 to 2023, and specifically a whole-system approach to quality assurance measures (Catholic Education Commission of Western Australia (CECWA) 2019).

### 3. Quality Catholic Education Western Australia

There is a unique governance structure in CEWA whereby the local Conference of Catholic Bishops collectively authorise Catholic education (Catholic Education Commission of Western Australia (CECWA) 2009). The Catholic Education Commission of Western Australia (CECWA) represents the diocesan bishops, and CEOWA serves as the secretariate for the Commission. Further to the unique governance structure, CEWA schools are accountable to the Minister for Education in WA and the Commonwealth Department of Education. Under the CEWA governance structure, system leaders have recently produced a quality assurance framework and supporting documentation called Quality Catholic Education (QCE). The work is a whole-system approach aligned to policy documents such as the Mandate for Catholic schools in WA (Catholic Education Commission of Western Australia (CECWA) 2009) and the Strategic Directions 2019–2023 (Catholic Education Commission of Western Australia (CECWA) 2019). QCE draws from the Catholic Church's mission to provide "a service which is truly civic and apostolic" (Congregation for Catholic Education (CCE) 1977, para. 4) and fulfils State and Commonwealth government registration and funding requirements aimed to improve the quality of learning opportunities for students (Department of Education 2022a).

QCE is one way the CEWA is demonstrating a commitment to the development of professional standards in education. For example, a review and improvement plan for Catholic schools is an outcome of QCE. As part of the review, the professional capabilities and standards of teaching and assessment practices by classroom teachers and school leaders, and in turn, the standards of student learning, are carefully evaluated. The professional standards developed by CEWA are drawn from work carried out by local, national and international educational authorities. The standards-setting work in CEWA is similar to initiatives in other jurisdictions, systems and sectors within WA (Department of Education 2022b), and across Australia (Department of Education 2022a).

In Australia, standards in teaching and assessment practices became apparent over a decade ago when evidence-based curriculum reforms spread through the educational landscape (Shaddock 2014). Similar educational reforms were evident in New Zealand, Canada, Ireland, Scotland, England and Wales, Japan and the United States of America. Much of these standards-setting reforms are aligned to accountability (Gardner 2012). At the time, State and Commonwealth governments introduced large-scale standardised assessments as one accountability measure to raise what they believed were declining standards in student learning. The focus of the concerns brought about certain initiatives including large-scale testing in literacy and numeracy (Australian Curriculum, Assessment and Reporting Authority (ACARA) 2022) and the development of professional standards for teachers and school leaders (Australian Institute for Teaching and School Leadership 2022). The educational reforms were and continue to be informed by specific and intentional assessment practices that encourage the collection of reliable and valid student achievement data using a range of well-tested measurement tools (Shaddock 2014). The purpose of such reforms is to raise cultural awareness of the standards of teaching and assessment practices and maximise ongoing improvements of those standards for the benefit of student learning (Department of Education and Training 2018; Masters 2018). Given the nature, role and prominence of the RE curriculum for CEWA schools, the QCE framework clearly outlines, in statement 2.2 Religious Education, how standards in RE are to be established and measured (Catholic Education Western Australia (CEWA) Limited 2022). The policy expectations in RE are no different to those of other learning areas.

## 4. Quality Assurance Measures in Religious Education

Within the CEWA QCE framework are policy and executive directives relevant to four categories or pillars: 1. Catholic identity, 2. Education, 3. Community and 4. Stewardship. The Education pillar contains the directives for RE, stating a commitment to deliver to "every student . . . high quality Religious Education. This requires active promotion and appropriate resourcing of Religious Education as the first learning area in every Catholic school, implementing the program promulgated by the diocesan bishop" (Catholic Education Western Australia (CEWA) Limited 2022).

Attempts to measure and review the quality of RE is no easy task and can prove challenging, especially when the use of the word 'quality' is not aligned to standards (Osterlind 2010). Furthermore, the nature and role of RE within the context of Catholic education is complex (Rossiter 2017; Scott 2015). Even though Catholic Church documents and local Catholic education policies such as the CEWA QCE framework highlight expected standards, the complexity surrounding RE is based on several competing factors related to religious and/or educational perceptions and consequential interpretations that require careful and practical curriculum considerations (McKinney and Sullivan 2013; Moran 2016; Poncini 2021). Such considerations include the collection and management of student achievement data using reliable and valid methods (Timperley et al. 2020). The student data are then used to generate resources that include scoping and planning strategies. For RE, the strategies have the potential to promote:

- Better connections and alignment between policy expectations and classroom and school practices;
- Improved awareness by classroom teachers and school leaders about the essential vocabulary and concepts of the RE curriculum;
- The integral nature of the components of the RE curriculum (i.e., content, pedagogy and assessment); and,
- Meaningful and purpose-filled assessment practices that recognise the significance of evaluating the quality of learning and the tools that attempt to measure that quality.

These considerations help classroom teachers and school leaders of RE to contextualise the teaching of the RE content and bring about intentionality to their teaching. The considerations acknowledge the significant links between CEWA's Christ-centred and student-focused vision for learning (Catholic Education Commission of Western Australia (CECWA) 2019; Catholic Education Western Australia (CEWA) Limited 2020d) and the origins and evaluative process underpinning the word assessment.

Assessment is an evaluative "process or processes of collecting and combining information from tasks (e.g., tests on performance or learning) with a view to making a judgement about a person or making a comparison against an established criterion" (Athanasou and Lamprianou 2002, p. 3). It comes from the Latin word *"assidere"* that means to 'sit beside'. Ideally, the use of the word assessment should take into account the "process of seeking and interpreting evidence for use by learners and their teachers, to identify where the learners are in their learning, where they need to go and how best to get there" (Gardner 2012, p. 3). The intention is to meet the differing needs of students from Kindergarten to Year 10, and specifically their learning in RE. As a result, positive changes to personal values, motivations, attitudes, behaviours and implied actions regarding the RE learning area may occur for teachers, and in turn, students (Poncini 2018, 2021).

Although the intention for rigorous teaching and assessment practices are considered and applied in Catholic education in Australia (Rymarz et al. 2021), the reality may be that teaching and assessment practices in RE are not as thorough as in other learning areas (Rossiter 2017). A reason for the lack of attention to formal assessment practices in RE may stem from the historical approaches to the teaching of RE based on the ongoing debates about the nature and role of Catechesis and Religious Education (Rossiter 2010; Scott 2015). Despite the debates, CEWA has developed various resources, including online lesson plans, assessment tasks and interactive activities to help classroom teachers and school leaders of RE focus on improving the quality of student learning in RE (Alteri et al. 2017).

To complement the existing pedagogical material and resources developed for the RE curriculum, other recent and not so recent, evidence-based standards-setting initiatives have been designed by the REFF Team at CEWA. One initiative used since 2007 to establish a measurable standard of the quality of student learning in RE is The Bishops' Religious Literacy Assessment (BRLA). The other educational initiatives include judging standards in RE, associated exemplars of achievement in RE and several reporting applications (Catholic Education Western Australia (CEWA) Limited 2020a, 2020b). Validity testing of the quality assurance measures is underpinned by research-based programs and projects. For example, the development and reporting of BRLA uses Rasch measurement theory (Andrich and Marais 2019) and the development of the judging standards is based on an Australia Research Council (ARC) Linkage Project LP180100046 conducted through the Australian Catholic University and supported by the University of Western Australia and other partners including CEWA (Adie 2022). Further to these initiatives and the methods used to evaluate them, other standards-based solutions are being considered to improve and progress student learning in RE. Such solutions include qualitative measures for better understanding students and their learning needs, targeted and differentiated support to students during the administration of assessments and tailored and adaptive testing. In keeping with QCE, the development and implementation of the proposed solutions would require collaboration between classroom teachers, school and system leaders.

### 4.1. The Bishops' Religious Literacy Assessment

Given the nature, role and prominence of the RE curriculum within CEWA, a large-scale, standardised assessment was introduced in 2007 to measure student learning in RE. The assessment is called The Bishops' Religious Literacy Assessment (BRLA). The assessment was developed to measure student knowledge and understanding of the content of the RE curriculum (Alteri et al. 2017). Students in Years Three (ages 8 to 9), Five (ages 10 to 11) and Nine (ages 14 to 15) in Catholic schools across all four Catholic dioceses in WA have participated in the assessment since it was first established. To date, over 200,000 students have completed the BRLA (Poncini 2021). The aims, general structure, and outcomes of the BRLA as a testing program for students in these three-year levels are modelled on large-scale assessment standards operating in Australian education (Australian Curriculum, Assessment and Reporting Authority (ACARA) 2022). Specifically, the BRLA adheres to the assessment principles and protocols outlined in the National Assessment Program: Literacy and Numeracy [NAPLAN]. For example, the general design, development, and implementation of the BRLA framework consists of five phases. These involve test item development, pre-administration, and administration processes, marking, analysis and reporting structures (Poncini 2018, 2021).

As part of the design framework, the BRLA undergoes reliability and validity screening on a yearly basis. Linking, equating, and scaling methodologies consistent with Item Response Theory (IRT) as a psychometric analysis of educational instruments are applied as part of the screening (Andrich and Marais 2019). For example, student responses to each of the test items are scrutinised using a particular IRT called Rasch, after the founder Georg William Rasch. A specific Rasch online program called RUMM2030 is used to evaluate test bias and differential item functioning. The analysis of test items from year to year determines the focus and style of future test items (Poncini 2018, 2021).

At the time of administration in August, students enrolled in Years Three, Five and Nine within most Catholic schools in WA are expected to complete the BRLA. The students complete a set of test items specific to their year level. Some of the test items are common to two and all three-year levels. The older the student cohort, the more test items provided, and the more time given to the students to complete the items. For example, Year Three students complete twenty multiple choice and ten short and extended response items in a 45-to-60 min timeframe. However, exemptions do apply, and special provisions are made for students who experience learning difficulties. That is, students who have trouble

reading can activate audio assist and students who have trouble writing can be assigned a scribe. Furthermore, parental consent is necessary for students to participate in the BRLA.

Following the administration of the BRLA, the students' responses to the multiple-choice and short answer response test items are automatically marked by specifically programmed computer technology (Poncini 2021). Classroom teachers and school leaders of RE have access to the students' scores instantaneously. The students' responses to the extended answer test items are marked online by a team of religious educators, employed by the CEOWA once the administration process is complete. The team of markers follow standardised protocols based on moderation principles involving pairwise comparisons (Heldsinger and Humphry 2010). For example, student responses are ranked and scored against a marking rubric. To achieve consistency in the scoring process, the marking rubric provides scoring information and exemplars of student performance. The results from the marking process are made available electronically to schools and parents in the form of individual student reports and class summary and item analysis reports.

Over time, the BRLA has produced longitudinal, baseline student achievement data at the classroom, school and system levels (Catholic Education Western Australia (CEWA) Limited 2020a). There is an expectation from CEOWA that the student achievement data generated from the BRLA are used by classroom teachers and school leaders to diagnose student learning in RE and consider future planning strategies that improve the quality of the learning. The expectation to analyse and interpret BRLA student performance data is standard practice for all large-scale, standardised assessments used by Catholic Education Western Australia (CEWA) Limited (2020a). Locally, nationally and internationally, classroom teachers and school leaders have been expected to be more data literate and confident in their analysis of student achievement data generated from these formal assessments (Shaddock 2014; Wyatt-Smith et al. 2019). This expectation applies to Catholic school educators (Holter et al. 2014), including religious educators (Rymarz et al. 2021).

The student achievement data from the BRLA are available on a CEWA reporting application (Catholic Education Western Australia (CEWA) Limited 2020a). The nature of the BRLA reporting application includes summary and comprehensive pages with longitudinal data, descriptive statistics specific to each year of administration and specifics relevant to item analysis. To support the whole-system approach to quality reporting practices, the CEOWA has developed the online reporting application to collect and manage student achievement data currently from standardised testing, including the BRLA. The reporting application is operated by Microsoft Office 365 Power BI technology (Microsoft Corporation 2022). Accessible on the reporting application also are student achievement data from NAPLAN, the Early Years Literacy and Numeracy data from the Observation Survey of Early Literacy Achievement (Clay 2019) and the Mathematics Assessment Interview and Associated Growth Point Framework (Clarke et al. 2002), and the Progressive Achievement Tests (Australian Council for Educational Research 2022).

Historically, the design of the BRLA was paper-based and the development of the BRLA was aligned to a large-scale testing program used by the WA government to measure the achievement of students in literacy, numeracy, science and society and environment. The assessment was called The Western Australian Monitoring Standards in Education (WAMSE) assessment program (School Curriculum and Standards Authority (SCSA) 2022b). The use of WAMSE and the BRLA signals a major change that occurred in education in the 1990s (in various countries around the world) to help improve the quality of student learning (Fullan and Quinn 2016). That change has led to evidence-based reforms in teaching and assessment practices. Like education systems around the world, CEWA is committed to improving the quality of student learning in all subjects. Therefore, the BRLA was developed and implemented in RE in response to the change in education.

In 2016, the BRLA was administered for the first time to Year Nine students online (Poncini 2021). By 2017, all students in Years Three, Five and Nine were administered the BRLA online. The online administration of the assessment is made possible by Creative Ministry Resources (CMR), an educational company providing liturgical and testing ser-

vices. CMR provides online testing in RE to Catholic education offices and dioceses in Australia and New Zealand. The educational company developed the online assessment platform over a three-year period in partnership with the Catholic Diocese of Parramatta (Creative Ministry Resources 2022).

Testing programs such as the BRLA can be traced back to the work of Scottish and Protestant scholars, Ninian Smart and Marie MacDonald (Poncini 2018). In Catholic circles, the National Catholic Education Association (NCEA) in the United States of America began developing assessments in RE for its parish-based catechesis programs. In Australia, the Catholic Education Office in Sydney was the first to implement a large-scale RE assessment consisting of multiple-choice test items. They engaged students in Years Six, Eight and Ten. The testing program continues today in other Catholic education offices and dioceses in Australia, such as Tasmania, South Australia, Wollongong, Maitland and Newcastle and Wilcannia-Forbes.

In accordance with Australian educational standards, teachers have a fundamental role in the quality of student learning and its success (Dinham 2016; Hattie 2009). Therefore, when it comes to the implementation of assessment practices, well-targeted professional development and learning for classroom teachers and school leaders is necessary (Gardner 2012; Shaddock 2014; Timperley et al. 2020). Given these expectations, religious educators should be held to the same standards as other educators and given the appropriate professional formation (Cullen 2019; Rymarz et al. 2021). For example, classroom teachers and school leaders of RE require support to analyse the BRLA student performance data. The analysis and interpretation processes should be objective and multidisciplined in their approach. They should consider the local Catholic school contexts and help teachers to make meaningful comparisons between the BRLA and other formal and informal in-class assessments. Classroom teachers and school leaders of RE should be encouraged and supported during professional development and learning opportunities to:

- Measure and assess students learning by focusing on growth as sustained improvement of the learning;
- Use data from the BRLA to deepen their understanding of the RE content;
- Better align the components of the RE curriculum and make explicit and implicit the teaching and assessment processes;
- Engage with the RE content through the application of skills (i.e., analysis and synthesis, making inferences, reflecting critically and problem solving); and,
- Collect reliable and valid student achievement data from their own in-class assessments.

The use of the BRLA over time has helped school- and system-based educators to measure the quality of student learning in RE. However, not all classroom teachers and school leaders of RE have agreed with its purpose and role in RE (Poncini 2018). This is because teacher perceptions of RE are complex and the use of a large-scale, standardised assessment in RE may not necessarily align with the differing approaches to teaching and assessment. Other measures of student learning such as the recently developed judging standards in RE have the potential to build on the work of the BRLA and improve the alignment between policy expectations with school-based practices in RE.

*4.2. The Judging Standards in Religious Education*

The judging standards in RE for Catholic schools in WA were first introduced in 2020 (Catholic Education Western Australia (CEWA) Limited 2020b). The structures and processes used to develop them are aligned to those used by SCSA to produce judging standards for all other learning areas implemented in government and non-government schools in WA (School Curriculum and Standards Authority (SCSA) 2022a). The format of the judging standards in RE is similar to the humanities subjects. Classroom teachers and school leaders can access the judging standards in RE via an internal CEWA intranet portal. The portal contains and manages other RE curriculum information as well as specific planning documents and pedagogical and assessment resources (Catholic Education Western Australia (CEWA) Limited 2020c).

The REFF Team, in consultation with various groups including classroom teachers, school leaders and learning area experts, developed the CEWA judging standards in RE. These documents present the expected progress of student learning in RE for year levels from Pre-Primary to Year 10, and as such, establish an appropriate standard for how RE is to be implemented within those year levels. The judging standards identify and summarise for each year level the essential vocabulary, knowledge and competencies or skills that students are expected to learn in RE. The judging standards also:

- Model the appropriate language and structures teachers can use for in-class assessments and year-level reporting; and,
- Assist teachers when reporting against the achievement standards to recognise differences in student learning for each year of schooling.

The judging standards in RE comprise three interrelated components: the achievement standard, the assessment pointers and annotated work samples.

1. The achievement standard draws directly from the RE curriculum and describes the expected level of achievement that most students should be working towards. The expected achievement for each year is described as 'C' or satisfactory. Some students are either progressing below or beyond the expected achievement standard. Students who are demonstrating achievement below the expected standard should receive additional support in their learning.
2. The assessment pointers exemplify evidence of skills and content in relation to the achievement standard. The pointers highlight what students may demonstrate rather than a checklist of everything students should know about the RE content. The pointers validate teachers' professional judgements when reporting against a five-point A to E scale and should be used with annotated work samples. Depending on what has been taught in the reporting period, teachers may refer to only a selection of the pointers in one or more of the contexts of the learning area.
3. The annotated work samples are collected from Catholic schools in WA to support teachers to report against the achievement standards. A collaborative effort involving classroom teachers, school leaders and CEWA consultants within teaching and learning directorates are involved in the collection process. To substantiate the work of the judging standards in RE, the first round of annotated work samples as exemplars of satisfactory achievement of student learning in RE for years Pre-Primary to Year 10 have been released. Other exemplars will follow. The annotated work samples help to identify and compare differences between student performance, relative to the achievement standards. They also encourage professional dialogue about student learning and the moderation of student learning, within schools and across the CEWA system.

The CEWA judging standards in RE share some of the features evident in standards-based documentation produced by educational sectors, jurisdictions, and systems, including those responsible for the RE learning area. The documentation uses measurable language to help teachers track student learning. For example, religious educators in Brisbane implement an RE curriculum whereby the online content is categorised according to descriptors and skill-based elaborations (Brisbane Catholic Education 2022). In another example, religious educators in South Australia refer to a RE curriculum linked to key capabilities continua that map religious awareness and practices (Catholic Education South Australia 2022). The key capabilities continua are linked to the curricula of other learning areas. In England, a framework with expectations, progression, and achievement in RE is developed for religious educators to use (Religious Education Council of England and Wales 2013). The framework is aligned to the frameworks established for other learning areas and outlines performance indicators of student learning and progress in RE.

Like the BRLA, the CEWA judging standards in RE can bring about focused intentionality to teaching, assessment and reporting practices in RE, as well as alignment between the components of the RE curriculum. The judging standards are the conduit between

the curriculum components of content, pedagogy, and assessment. As such, they support teachers to make judgements about student learning based on an evaluation of their own teaching and assessment practices against a set of specific standards and against a contextual understanding of the content. As an educational quality assurance measure, they provide common specifications for supporting dialogue and practices aimed at progressing student learning. This is because they present a set of criteria and information to help teachers make better sense of the content in the curriculum derived from the Catechism of the Catholic Church. That is, they highlight and attempt to address the fundamental and common questions that arise from the content: "Who are we as Catholics and what is our mission?" "What do we believe?" and "How do we practice?".

The judging standards in RE are a by-product of research (Catholic Education Western Australia (CEWA) Limited 2020b). The development of the judging standards is part of a broader research project involving CEWA, in partnership with the Australian Catholic University (ACU) and the University of Western Australia (UWA). The research is an Australian Research Council (ARC) Linkage Project LP180100046 entitled, 'Improving teacher assessment capabilities using exemplars and online moderation' (Adie 2022). The research focuses on student learning in Years Four, Six and Eight and the disciplines of English, Mathematics, Science and RE. The aim of the research is to investigate and understand how teachers make judgements about student learning. The investigation uses scaled student work samples exemplifying a five-point letter A to E grading system or equivalent descriptors. The grading system is common to Australian school-based education for students enrolled in Pre-Primary to Year 10 and fulfils the reporting requirements for all Australian schools. The research includes annotated cognitive commentaries written by teachers, showing how their judgements are made, and potentially support the comparability of judgements. The benefits of the research enhance assessment design and the use of assessment evidence, as well as improving teachers' focus on content knowledge and instructional decision making. Professor Lenore Adie from the Institute for Learning Sciences and Teacher Education at ACU is the chief investigator of the research. Associate Professor Stephen Humphry from UWA is a leading expert in educational psychometrics. Professor Humphry has contributed extensively to the project, and specifically to establishing standards-setting resources in RE, such as the BRLA and the judging standards.

Aligned to the research project, and to further promote moderation and progression of student learning in RE (within schools and across the CEWA system), a psychometric scale with related calibrated exemplars of student learning in RE is being developed by CEWA. The scale is equivalent to the BrightPath Teacher's Ruler for writing and mathematics assessments (Pairwise 2022). This scale will present statistical and qualitative information about students' progression in learning in RE and will help classroom teachers and school leaders of RE to moderate student learning generated from in-class assessments against the scaled exemplars. The moderation opportunities will pave the way for reviewing and refining set standards in RE and the learning interventions that are planned. If these moderation opportunities are provided to religious educators responsible for students enrolled in the early years of schooling (e.g., Pre-Primary to Year Two), the greater the progression and growth of student learning in RE.

Regardless of the potential benefits of the BRLA and the judging standards in RE, classroom teachers and school leaders of RE at the school and system levels need to recognise and apply both the religious and educational dimensions underpinning the RE learning area. These dimensions are very much a part of the whole Catholic school curriculum (D'Orsa and D'Orsa and D'Orsa 2012). As trained professionals in Catholic schools, classroom teachers and school leaders of RE need to ensure a certain quality of student learning in RE and the success of that learning. They need to understand and methodically deliver the content of the RE curriculum following contemporary educational standards that adhere to pedagogical, assessment and reporting designs and practices, as outlined for the other learning areas (Australian Curriculum, Assessment and Reporting Authority (ACARA) 2022; School Curriculum and Standards Authority (SCSA) 2022a). Similarly, CEWA and

academic institutions such as The Catholic Institute of Western Australia (CIWA) and The University of Notre Dame Australia (UNDA) have a responsibility for providing professional, spiritual, religious, and educational formation to classroom teachers and school leaders of RE that in turn improve the quality of student learning.

To help support the institutions that provide religious educators with ongoing professional and faith formation, collaboration and targeted research is needed (Poncini 2018). Collectively, all educators involved in the development and implementation of the RE curriculum have a significant role to play in improving and progressing student learning. They are called to interpret the RE curriculum in holistic ways that promote the integration of faith, life and culture for students and educators alike (D'Orsa and D'Orsa and D'Orsa 2020; Rymarz and Franchi 2019). Future studies that investigate student and teacher perceptions of the educational initiatives described in this paper are necessary. Such research may reveal the true benefits of the educational initiative if any, and how best to improve and progress student learning in RE.

## 5. Conclusions

McKinney writes, "A Catholic understanding of education can have a dual purpose. It provides an understanding of Catholic education and a Catholic understanding of education per se" (McKinney 2021, p. 191). This paper attempts to address the dual purpose of: (i) presenting educational initiatives that are evidence-based and specific to the RE learning area, reserved for classrooms in Catholic schools in Western Australia and (ii) an understanding of standards-setting, quality assurance measures adopted in RE but not unique to RE. The dual purpose of this paper has been to describe educational initiatives that may serve as examples aimed to help classroom teachers and school leaders address the quality of student learning, regardless of the learning area.

The initiatives in this paper have focused on the quality of student learning and include a large-scale standardised assessment called the BRLA, judging standards, exemplars of student achievement and reporting applications. These are quality assurance measures aligned to a whole-system approach to teaching and assessment practices typical of other learning areas but not always evident in RE. The measures are relevant to the Western Australian Kindergarten to Year 10 curriculum. Collaborative efforts by school- and system-based educators to engage with the advancement of student learning in RE have helped develop and implement the initiatives. The outcomes are moderation and evaluative opportunities that align and benefit teaching and assessment practices. Furthermore, the opportunities may inspire dialogue that can foster a better understanding of the evangelising mission of the Church by promoting improved religious literacy, competency, understanding, appreciation, and application of the Catholic Faith Tradition.

**Funding:** This paper received no external funding.

**Conflicts of Interest:** The author declares no conflict of interest.

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
