# Peer review of "Standards Setting in Religious Education: Addressing the Quality of Teaching and Assessment Practices"

_religions, doi:10.3390/rel14030315_

Round 1

Reviewer 1 Report

The issue of quality assurance measures in the field of education is always of great importance.

The summary of the study is in a good structure. However, keywords should be arranged so that each of them consists of no more than two words. The problem situation of the study is explained in a good structure with current sources. The subjects of the study can be given as objectives under the introduction part and the subjects can be presented as sub-titles in the results part. The biggest problem of the study is that the working method is not clearly stated. It should be explained by adding the method part and the results part should be given.

The conclusion part is sufficient. Detailed recommendations can be made for future research.

Author Response

Thank you so much for reviewing my paper entitled, Standards setting in Religious Education: Addressing the quality of teaching and assessment practices.

I understand your comments regarding the key words and have tried to rectify the issue by reducing the key words to either one or two words combined.

In terms of your other comments, my paper is not a study, it is a presentation of educational initiatives: (i) developed specifically for the RE learning area in Catholic Education Western Australia and (ii) aligned to other learning areas implemented in Western Australian Kindergarten to Year 10 classrooms.

Given that the paper is not a study, there are no research questions, nor considerations for research design and methodology. Hence, no findings are reported in this paper.

The aim of the paper is only to describe the educational initiatives and identify how the initiatives support local and national policy expectations relating to the treatment of RE as a learning area taught in Catholic schools. The paper identifies how the described, educational initiatives also build on local, national and international scholarship in education that relates to the quality and progression of student learning.

The significance of the RE learning area is identified within the paper and the justification for the evidence-based, educational initiatives is provided. For example, the educational initiatives may benefit the RE learning area by: (i) supporting the teaching and assessment practices of religious educators, (ii) providing comparative standards of student learning and (iii) encouraging a whole-system approach to improving the quality of student learning in RE. Furthermore, on page 2, I provide an explanation for why the educational initiatives may benefit the RE learning area from an Australian Catholic perspective.  That is, “At the local level, the educational initiatives described in this paper build on the whole of CEWA priority for Quality Catholic Education (QCE) and the professional development and formation of classroom teachers and school leaders of RE as part of that priority (Catholic Education Western Australia [CEWA], 2022a). Beyond the local level, the initiatives promote the on-going dialogue and collaboration needed between classroom and school and system leaders about teaching and assessment practices in RE that focus on the quality of student learning (National Catholic Education Commission [NCEC], 2018).”

Given the paper only describes the educational initiatives, a study of stakeholder perceptions of the initiatives is for future research. A study of the implications for using the educational initiatives is necessary and worthwhile and is being considered. On page 8, of the paper, I have included findings from my own research about the long-standing initiative as the large-scale standardised assessment in RE and have proposed the need for future studies that investigate the use of the other educational initiatives described in this paper, such as the judging standards in RE. On page 10, I reaffirmed the need for future research specific to investigating teacher perceptions of the educational initiatives.

I look forward to any assistance you may provide that specifically identifies on the paper where changes need to be made.

Reviewer 2 Report

I honestly did not understand the purpose of the article.

At present, it is a description of an educational system, standards, and some evaluative tools. It was unclear to me the purpose of the paper beyond description.  If that was the purpose and the editors wanted to include it for that purpose, it is acceptable.  

However, I think the author is attempting to argue that RE is important, and can be improved significantly by evidence-based research.  If this is the case, the paper does not adequately describe a problem or issue and does not make an argument.  I think it might be useful information, if the authors were clear about the purpose and impact of the article and then rewrite.

At present the mere description does not seem to move to conclusion or implications for practice. 

Author Response

Thank you so much for reviewing my paper entitled, Standards setting in Religious Education: Addressing the quality of teaching and assessment practices.

I understand your comments regarding the purpose of the paper and want to highlight that my paper is not a study, it is a presentation of educational initiatives: (i) developed specifically for the RE learning area in Catholic Education Western Australia and (ii) aligned to other learning areas implemented in Western Australian Kindergarten to Year 10 classrooms.

Given that the paper is not a study, there are no research questions, nor considerations for research design and methodology. Hence, no findings are reported in this paper.

The aim of the paper is only to describe the educational initiatives and identify how the initiatives support local and national policy expectations relating to the treatment of RE as a learning area taught in Catholic schools. The paper identifies how the described, educational initiatives also build on local, national and international scholarship in education that relates to the quality and progression of student learning.

The significance of the RE learning area is identified within the paper and the justification for the evidence-based, educational initiatives is provided. For example, the educational initiatives may benefit the RE learning area by: (i) supporting the teaching and assessment practices of religious educators, (ii) providing comparative standards of student learning and (iii) encouraging a whole-system approach to improving the quality of student learning in RE. Furthermore, on page 2, I provide an explanation for why the educational initiatives may benefit the RE learning area from an Australian Catholic perspective.  That is, “At the local level, the educational initiatives described in this paper build on the whole of CEWA priority for Quality Catholic Education (QCE) and the professional development and formation of classroom teachers and school leaders of RE as part of that priority (Catholic Education Western Australia [CEWA], 2022a). Beyond the local level, the initiatives promote the on-going dialogue and collaboration needed between classroom and school and system leaders about teaching and assessment practices in RE that focus on the quality of student learning (National Catholic Education Commission [NCEC], 2018).”

Given the paper only describes the educational initiatives, a study of stakeholder perceptions of the initiatives is for future research. A study of the implications for using the educational initiatives is necessary and worthwhile and is being considered. On page 8, of the paper, I have included findings from my own research about the long-standing initiative as the large-scale standardised assessment in RE and have proposed the need for future studies that investigate the use of the other educational initiatives described in this paper, such as the judging standards in RE. On page 10, I reaffirmed the need for future research specific to investigating teacher perceptions of the educational initiatives.

I look forward to any assistance you may provide that specifically identifies on the paper where changes need to be made.

Reviewer 3 Report

This draft has is some interesting information about assessment practices in some Australian Catholic schools. It provides a reasonable overview of such ways of working for those who are not part of the scene it describes. 

There are some questions about the aim of the paper: what is its actual argument? It seems mostly to be a descriptive account with little by way of critique. Perhaps some of the descriptive material could be summarised to allow space for a critical appreciation of the ideas presented.

Some specific points now follow:

1.     In the Introduction explain what the purpose of the paper is. At present it is not too clear.

2.     The Introduction needs to clarify terms like ‘educational initiatives.’ Lines 25-26 need to be more specific.

3.     There is also a need to clarify (in the Introduction) the possible dissonance between the  ‘assessment practices outside the scope of traditional and confessional RE classrooms’ (lines 30-31) and an in-house assessment (BRLA) on religious literacy (line 35).

4.     Say more about the relationship between evangelisation and RE (lines 50 and 79-80). The reference to the Directory for Catechesis (PDPNE 2020) is important but the Directory is a very subtle and nuanced document when it speaks of school-based RE (see paras. 313-318).

5.     Paragraph 2 in section 2 (lines 71-88) needs to breathe a little more: things are very tightly packed. Identify the relationship (and possible tensions) between school-based RE and catechesis

6.     Ponder more deeply the relationship between RE and the wider curriculum (lines 101-105).

7.     In sections 3 and 4 it might be worth considering shorter descriptions and raising questions about the extent to which the initiatives mentioned are successful - and what is understood by success in this field.

8.     4.1 in particular could be more succinct and thus allow space for a critical reflection on the effectiveness of the BRLA as an instrument.

9.  If the aim of the paper is clearly stated at the start, this would allow the writer to make some initial judgements /evaluations. The comments made in lines 497-501 are not really supported by the paper and come over more as assertions than conclusions. If there is evidence in support of such conclusions we need to see what it is.

The paper is potentially important for Catholic schools, and not just in WA. There is a need to make a reasoned evaluation of assessment practices across the curriculum, not just in RE. In some places RE has fallen behind other curricular domains in terms of assessment and critical thinking about its operation. Papers such as this (duly revised) offer some important lights for the path ahead.  

Author Response

Thank you very much for taking the time to review my paper. I greatly appreciate your thorough and specific feedback.

My paper is not a study. Instead, the paper is a presentation of educational initiatives specific to the RE learning area but not unique to RE. Within the introduction of the paper, I identify the initiatives as quality assurance measures. The initiatives described in this paper are: (i) developed for the RE learning area offered within Catholic Education Western Australia; (ii) aligned to the practices evident in other learning areas that are implemented in Western Australian Kindergarten to Year 10 classrooms; and, (iii) endorsing a Quality Catholic Education policy for CEWA. Therefore, I believe that I have adequately addressed the nature of the educational initiatives throughout the paper. 

Given the paper is not a study, content regarding research questions, design, methodology and findings are not applicable. However, the paper does endorse future research to further enhance the development of the educational initiatives and follow-up measures. For example, refer to the paragraph from line 491 and the final comments in the conclusion where I reaffirmed the need for future research investigating the success of the initiatives by specifically investigating student and teacher perceptions of the described initiatives. Also, I have included on pages 8 and 10 relevant findings from my own research regarding teacher perceptions of the BRLA.

This paper is limited in scope and content to the complexities surrounding RE within the context of Catholic education and the evangelising mission of the Catholic Church. Also, I am sorry, but I am unsure of the improvements/additional citations and references relevant to my topic. I have referenced the Directory, but more so, key Vatican documents such as those from the Congregation of Catholic Education, that over the years have been the primary source of information specific to Religious Education.

Even though this paper may not appear to answer the issues you raise in points 3 to 6, I believe Chapters 2 and 3 together with the findings from my thesis will. For example, in Chapter 2, I explain the relationship between RE and Catechesis. And, the findings from my study do address the disconnect between policy expectations and school-based practices in RE. You might be interested in reading my thesis entitled, Teacher Perceptions of a large-scale testing in Religious Education: How do classroom teachers and school leaders perceive the Bishops' Religious Literacy Assessment?  

Despite the intended scope of this paper, I have made several changes to the paper in response to your feedback. The changes reflect a specific argument for why the paper describes the educational initiatives and how the initiatives may impact in a positive way to student learning in the early years in RE and beyond. I have also attended to all spelling and punctuation errors. I have continued to use UK English and not US English.

I have reviewed the introduction and conclusion of the paper. In doing so, I have added paragraphs to each that are specific to the purpose and benefits of the paper. Extra sentences and whole paragraphs have been inserted into the paper to affirm the purpose of the paper. The purpose is linked to the benefits gained from the educational initiatives. For example:

  1. Throughout the paper, I have inserted clarifying words to ensure a better link between ideas and the central argument for writing the paper. 
  2. From line 6, the abstract states that the “paper describes a set of educational initiatives …”
  3. From line 27, two new paragraphs have been added to explain the purpose and benefits of the initiatives as described in the introduction.
  4. From line 515, the conclusion has been re-shaped to re-affirm the purpose and benefits of the educational initiatives described in the paper.

Again, thank you for your feedback. I look forward to your response. 

Reviewer 4 Report

Thank you for a comprehensive account of the complex system of assessment for RE in CEWA.  It provides the non-Australian reader with sufficient background, context and information to understand the benefits of this article for the corpus of literature not only for assessment practice but promoting the formalised recognition of such as essential in Catholic Religious Education.

The following a request only and is not a required amendment to the article which I would consider publishable in its current state (minus the tracked changes still visible).  During your research on this topic, what, if any, are areas of achievement that are not currently met by the existing measures as described?

Author Response

Thank you very much for taking the time to review my paper entitled, Standards setting in Religious Education: Addressing the quality of teaching and assessment practices.

I have reviewed the paper and attended to any spelling and punctuation errors. I have, however continued to use UK English rather than US English.

I have also taken on your advice and decided to actually add extra clarifying sentences to identify potential solutions for addressing student achievement that is not currently met by the described initiatives. For example,

  1. from 261 an elaboration of future initiatives is outlined that answers the question of what areas of achievement are not being currently met by the existing measures as described.
  2. from line 493 adaptive testing is acknowledge to better meet the individual needs of students and capture contextual information from students and their learning
  3. from line 547 in the conclusion whereby areas of achievement such as student knowledge and understanding of

The solutions address the known limitations to traditional test item development such as the reliability and validity of multiple-choice items and the type of differentiated support that should be provided to students who experience difficulties reading and writing. Thank you again for your question. 

Round 2

Reviewer 1 Report

After the corrections, the article is of a quality that can be published.

Author Response

Thank you for your valuable feedback. 

I greatly appreciate your time and support. 

Reviewer 2 Report

I believe the essay needs a thesis an argument. At present it is a mere descriptions. A robust essay would have a thesis about the value of evaluation, make an argument, and show its contribution to teaching/learning. 

Author Response

Thank you very much for taking the time to review my paper again. I greatly appreciate your specific feedback.

I have made further changes to the paper. The changes reflect a specific argument for why the paper describes the educational initiatives and how the initiatives may impact in a positive way to student learning in the early years in RE and beyond. I have also attended to all spelling and punctuation errors. Please be aware that I have continued to use UK English and not US English.

I have reviewed the introduction and conclusion of the paper. In doing so, I have added paragraphs to each that are specific to the purpose and benefits of the paper. Extra sentences and whole paragraphs have been inserted into the paper to affirm the purpose of the paper. The purpose is linked to the benefits gained from the educational initiatives. For example:

  1. Throughout the paper, I have inserted clarifying words to ensure a better link between ideas and the central argument for writing the paper. 
  2. From line 6, the abstract states that the “paper describes a set of educational initiatives …”
  3. From line 27, two new paragraphs have been added to explain the purpose and benefits of the initiatives described in the paper.
  4. From line 515, the conclusion of the paper has been re-shaped to re-affirm the purpose and benefits of the educational initiatives described in the paper.

Again, thank you for your feedback. I look forward to your response. 

Reviewer 3 Report

This is much improved. I suggest that the title would be improved by including the word 'study' somewhere. This would give focus to what follows.

Sort out the reference for McKinney on p. 11, line 531. It should be McKinney in Convery, Franchi and Valero and then put the full title in the reference list.

Author Response

Thank you again for your time.

I greatly appreciate your comments. 

I believe I have addressed the citation and referencing issue. I have also reviewed my reference list and identified errors that needed to be corrected. 

As my paper is a description and not a study, I'm not sure what is meant by your comment and have not made a change to the title of my paper. 

God bless

Round 3

Reviewer 2 Report

I am happy to approve for publication.  The author is clear about purpose — description.  I think it will be on use to help show what countries are doing and how it affects research education.  I hope the author moves forward to explore evaluation standards and their effectiveness. 

Author Response

Thank you again for your comments. 

I have added a paragraph (lines 447 to 460) about judging standards and specifically examples of similar standards-based approaches in RE. The examples are from Catholic education in Brisbane and South Australia. An example from England is also provided.

Citations and referencing are included. 

I greatly appreciate your comments and I do look forward to future research.

God bless